**Data Availability Statement:** The sequences were deposited into GenBank with accession numbers MW275077-MW275119.

# An epidemiological surveillance of hand foot and mouth disease in paediatric patients and in community: A Singapore retrospective cohort study, 2013–2018

**Nyo Min**[1‡], **Yasmin Hui Binn Ong**[1‡], **Alvin X. Han**[2], **Si Xian Ho**[1], **Emmerie Wong Phaik Yen**[3], **Kenneth Hon Kim Ban**[4], **Sebastian Maurer-Stroh**[2,5], **Chia Yin Chong**[3], **Justin Jang Hann Chu**[1,6]*

1 Laboratory of Molecular RNA Virology and Antiviral Strategies, Department of Microbiology and Immunology, Yong Loo Lin School of Medicine, National University of Singapore, Singapore, Singapore, 2 Protein Sequence Analysis Group, Bioinformatics Institute, Agency for Science, Technology and Research (A*STAR), Singapore, Singapore, 3 Infectious Disease Service, Department of Pediatrics, KK Women's and Children's Hospital, Singapore, Singapore, 4 Department of Biochemistry, Yong Loo Lin School of Medicine, National University of Singapore, Singapore, Singapore, 5 Department of Biological Sciences (DBS), National University of Singapore (NUS), Singapore, Singapore, Singapore, 6 Collaborative and Translation Unit for HFMD, Institute of Molecular and Cell Biology, Agency for Science, Technology and Research (A*STAR), Singapore, Singapore

‡ These first authors contributed equally to this article.
* miccjh@nus.edu.sg

## Abstract

### Background

While hand, foot and mouth disease (HFMD) is primarily self-resolving—soaring incidence rate of symptomatic HFMD effectuates economic burden in the Asia-Pacific region. Singapore has seen a conspicuous rise in the number of HFMD cases from 2010s. Here, we aims to identify the serology and genotypes responsible for such outbreaks in hospitals and childcare facilities.

### Methods

We studied symptomatic paediatric HFMD cases from 2013 to 2018 in Singapore. Surveillance for subclinical enterovirus infections was also performed in childcares at the same time period.

### Results

Genotyping 101 symptomatic HFMD samples revealed CV-A6 as the major etiological agent for recent outbreaks. We detected infections with CV-A6 (41.0%), EV-A71 (7%), CV-A16 (3.0%), coxsackievirus A2, CV-A2 (1.0%) and coxsackievirus A10, CV-A10 (1.0%). Phylogenetic analysis of local CV-A6 strains revealed a high level of heterogeneity compared against others worldwide, dissimilar to other HFMD causative enteroviruses for which the dominant strains and genotypes are highly region specific. We detected sub-clinical

**Funding:** This study was supported by National Medical Research Council, NMRC/CBRG/0059/2014 to JJHC (https://www.nmrc.gov.sg/) and Ministry of Education - Singapore (SG) MOE2017-T2-1-078 to JJHC and Ministry of Education Singapore; MOE2017-T2-2-014 to JJHC. (https://www.moe.gov.sg/) The funders have played no role in the study design, data collection and analysis, decision to publish, or preparation of the manuscript.

**Competing interests:** The authors have declared that no competing interests exist.

enterovirus infections in childcare centres; 17.1% (n = 245) tested positive for enterovirus in saliva, without HFMD indicative symptoms at the point of sample collection.

## Conclusions

CV-A6 remained as the dominant HFMD causative strain in Singapore. Silent subclinical enteroviral infections were detected and warrant further investigations.

## Author summary

In most cases, Hand Foot and Mouth Disease or HFMD typically manifest in mild fever along with sore throat and rashes on the body. From 2010 onwards, Singapore has seen a steady increase in the case number of HFMD reaching tens of thousands in recent years. HFMD is caused by intestinal viruses and in this study, we established with molecular surveillance methods that one of the causative serotypes, CV-A6 is the major etiological agent for HFMD in Singapore for the current decade. We discovered that circulating enterovirus, CV-A6 in Singapore share similarities in genetic make-up to those currently circulating strains found worldwide and found to be especially close to the ones in neighbouring countries. HFMD spreads from person to person, especially in high-risk areas such as childcare centers where children congregate. Therefore, we conducted saliva collections routinely from childcare centers across Singapore and found that subclinical enterovirus infections have also been prevailing in clusters, occurring silently and unnoticed.

## Background

Hand, foot and mouth disease (HFMD) primarily affects children below the age of five and was one of the top 5 most contagious febrile viral illness of 2018 in Singapore [1,2]. Infection with more than 20 genetically diverse viruses from the genus *Enterovirus*—family *Picornaviridae* causes HFMD which consequently impedes vaccine and antiviral development despite half a century old endemic status in majority of Asia-Pacific region [2–4]. Albeit serotype dominance of HFMD largely depending on temporal and geographical factors, underlying epidemiologic indices inflicting such paradigms are still not clearly understood [5]. Although human Enterovirus A71 (HEV-A71) and Coxsackievirus A16 (CV-A16) were major causative agents of HFMD in Singapore from 2000 to 2010, severely limited epidemiological surveillance data is available from 2010 forward despite several large outbreaks [2,6,7].

Multiple EV-A71 vaccine has been approved for use in China particularly against subgenotype C4a, which decently protected participants in the intervention arm against severe HFMD with a good safety profile [5,8,9]. Regardless, the lack of cross protection against Coxsackievirus group handicapped the efficiency of the vaccine as the coxsackieviruses became dominant etiological agent causing HFMD over the last decade in several countries [3,10,11]. Since Enteroviruses are RNA viruses prone to mutation with every replication cycle conferring better survival fitness, constant molecular surveillance is critical to understand the ever-changing viral phylodynamics and their impact on HFMD transmission. In this study, we attempted to type the various HEVs present in clinical specimens, taking a glimpse into the current HFMD situation in Singapore from the period of 2013 to 2018. Interestingly, despite being the major etiological agent for recent outbreaks across several countries, there is no general consensus on

sub-genotype of CV-A6 unlike EV-A71 and CV-A16. Here, we employed phylogenetic methods to analyse for genetic changes in antigenic viral VP1 region throughout the world for CV-A6. Additionally, we also compared the diagnosis efficacy between the use of traditional throat swab samples with saliva samples. This is to analyse if saliva could be used as a possible diagnostic medium instead of throat swab as the collection process of latter triggers the gag reflex causing nausea. Next, this study also attempts to detect silent subclinical enterovirus infections in Singapore childcare centres; little is known about the subclinical cases in Singapore and HFMD outbreaks often occurs sporadically and frequently among these institutions.

Above all, epidemiological surveillance is a paramount instrument in anticipating and minimising outbreaks [12]. By understanding the circulating serotypes, genotypes and sub-genotypes along with their evolutions in paediatric patients and asymptomatic individuals, we aim to better appreciate HFMD transmission patterns, corresponding to their respective clinical outcomes which paints a holistic picture that captures HFMD causing human enterovirus epidemiology in Singapore.

## Methods

### Ethics statement

This study was approved by the centralised institutional review board (CIRB) of Singhealth under CRIB number 2012/448/E for paediatric samples. National University of Singapore (NUS) IRB granted routine collection of childcare samples under the protocol, B/14/273. As all participants are minors under the age of 21 in this study, **the formal consent was obtained from the parent or guardian of the participant.**

### Patient cohorts

From Kandang Kerbau (KK) Women's & Children's Hospital, 104 Saliva and throat swab biospecimens were collected from paediatric patients admitted for symptomatic HFMD diagnosed by clinicians according to WHO guideline between June 2013 to January 2018. From approved childcare centres, 245 saliva samples were obtained from healthy volunteers void of symptoms indicative of HFMD such as fever; rashes on palm, hand and soles of feet; ulcer in the mouth and blisters at the point of collection. Informed consent was obtained from parents and/or guardians of healthy volunteers prior to the collection.

### Detection of enteroviruses

Throat swab samples collected from KK Hospital were transferred into the virus transport medium. All saliva samples were collected using SalivaBio Children's swab (Salimetric Inc., Carlsbad, California). Viral RNA extraction was done using Qiagen mini viral RNA extraction kit (Qiagen, California, USA) according to manufacturer's protocol. The viral RNA was then reverse transcribed and amplified. For pan-enterovirus detection, primers were adapted from a previous study which flanks highly conserved regions of the 5'Untranslated region (5'UTR) permitting broad spectrum detection [13]. For Enterovirus A serotyping, degenerate primers which primes the hypervariable VP1 region of the viral genome was utilized; PCR fragments were gel extracted and sequenced using Sanger sequencing method [14]. Enterovirus typing tool from Genome Detective workflow was used to identify the serotype, genotype and sub-genotype of sequenced enteroviruses [15].

For NGS library preparation, extracted total RNA obtained from saliva sample were fragmented and cDNA synthesis was performed by using Maxima H Minu Double-Stranded cDNA synthesis kit in accordance to manufacturer's protocol (Thermo Fisher Scientific,

Massachusetts, USA). Library preparations were performed by following the SeqCap EZ HyperCap Workflow with KAPA Hyper Prep Kit and NimbleGen adapter kits (Roche Diagnosistics, Risch-Rotkreuz, Switzerland). Hi-seq 4000 workflow was used for NGS sequencing and sequences obtained were demultiplexed using Illumina software. Resulting sequences were processed using Genome Detective online classification tool [15].

### Global CV-A6 sequence curation and phylogenetic analysis

5,432 CV-A6 VP1 reference sequences with country and collection date information were downloaded from GenBank. After removing identical nucleotide sequences using CD-HIT [16], the remaining sequences were aligned using MAFFT [17] and trimmed to the same length as the Singaporean sequences. In total, there were 2,397 reference sequences left. As most of the sequences were collected in China (~70%), the alignment was randomly down-sampled to 1,158 reference sequences, ensuring an equitable representation of sequences from every country in every calendar year.

Maximum likelihood phylogenetic trees were reconstructed using RAxML (v8.2.12; General reversible time substitution model with gamma-distributed rate heterogeneity) [18] from former nucleotide alignments. Confidence of the internal nodes were estimated based on 1000 bootstrap trees.

## Results

One hundred and four paediatrics patients were enrolled in the study from the period of 2013 to 2018 from KK Women's & Children's Hospital, of which 92.3% (n = 96) were of 5 years and below and comparatively there are more males admitted than females (**Table 1**). Approximately a quarter of the patients reported to have pre-existing medical conditions, such as anaemia, eczema or other infections (**Table 1**). Additionally, half of the patients recalled to be being exposed to individuals who were sick prior to their admission into the hospital (**Table 1**). The hospitalisation duration ranged from 0 to 12 days, with a mean and median of 2.8 and 2.0 days respectively and 43.3% of the patients were placed on drips during their hospitalisation due to dehydration (**Table 1**). There were no fatal or severe cases of HFMD reported with 92.3% (n = 96) having a full recovery and 7.7% (n = 8) were discharged with partial recovery status, in which the patient is discharged with either one or more WHO defined HFMD symptom. Commonly observed symptoms includes fever (90.4%), mouth ulcer (89.4%), skin lesions (89.4%), rashes (81.7%) and papules (80.8%) (**S1 Fig**). The average maximum body temperature at day 1 of admission was 38.3°C. A significant portion of the patients (24.0%; n = 25) reported being poorly fed, likely exacerbated by painful ulcers in the buccal cavity that made oral intake excruciating, subsequently causing lethargy in these patients. Other atypical clinical signs recorded include breathlessness, throat and abdominal pains, albeit at lower frequencies (**S1 Fig**).

The efficacy of throat-swab and saliva samples in detecting enteroviruses was examined using pan-entero RT-PCR detection assay; 31 patient samples were sub-sampled; enteroviruses were detected in 27 throat swab samples and 21 corresponding saliva samples. The calculated sensitivity for saliva and throat swab samples was 71% and 87% respectively (**S1 Table**). Evidently, throat swab sample was evidently more efficient in detecting enteroviruses, possibly due to the ability of HEVs to replicate in tonsillar cells [19] and the use of virus transport medium to preserve the virus after the collection of the throat swab samples. Viral load in several samples might have fallen below the detection threshold of the assay, hence, consequentially undetected.

**Table 1. Sociodemographic distribution of pediatric patients recruited into this study.** Characteristics collected from pediatric patients admitted for HFMD into KK Women's & Children's Hospital includes their age, gender, race, hospitalization duration, contact history, IV drip status and other parameters.

| Clinical (N = 104) | |
|---|---|
| **Characteristics** | **Frequency (%)** |
| **Age** | |
| ≤5 years | 96 (92.3) |
| >5 years | 8 (7.7) |
| **Gender** | |
| Female | 37 (35.6) |
| Male | 67 (64.4) |
| **Race** | |
| Chinese | 49 (47.1) |
| Malay | 45 (43.3) |
| Indian | 9 (8.6) |
| Others | 1 (1.0) |
| **Other disease cohort characteristics** | |
| Underlying conditions | 26 (25.0) |
| Anemia | 6 (5.8) |
| Eczema | 6 (5.8) |
| Other infections | 9 (8.6) |
| Others | 5 (4.8) |
| **Contact history** | |
| Yes | 51 (49.0) |
| Family members | 35 (33.7) |
| Others | 5 (4.8) |
| Unchecked | 11 (10.6) |
| No | 53 (51.0) |
| **Hospitalisation duration (days)** | |
| Minimum | 0.0 |
| Maximum | 12.0 |
| Median | 2.0 |
| Average | 2.8 |
| **IV drip status** | |
| Positive | 45 (43.3) |
| Negative | 59 (56.7) |

One hundred and one patients successfully completed the study and throat swab samples were subjected to pan-entero RT-PCR workflow as described previously; 81% (n = 82) were detected with the presence of enterovirus RNA (**Fig 1A**). Degenerated primers targeting the hypervariable partial VP1 region of the viral genome is used for genotyping by Sanger sequencing and 46% (n = 46) of the samples were successfully typed to enteroviruses (**Fig 1A**). Full length NGS of viral RNA from saliva agreed with Sanger sequencing on previously typed CV-A6 samples (n = 8); samples that were untypeable were revealed to be CV-A6 (n = 5), EV-A71 (n = 1) and CV-A10 (n = 1) (**Fig 1A**). It is noteworthy that majority of the samples that were successfully typed belongs to Coxsackievirus A6 (CV-A6) while only 1 sample was typed to CV-A2, which might have co-circulated alongside EV-A71 and CV-A16 in Singapore (**Fig 1A**). With VP1 sequencing, 34% (n = 34) of the samples were not successfully typed but were positive when detected for HEVs with pan-entero RT-PCR test (**Fig 1A**). Noticeably,

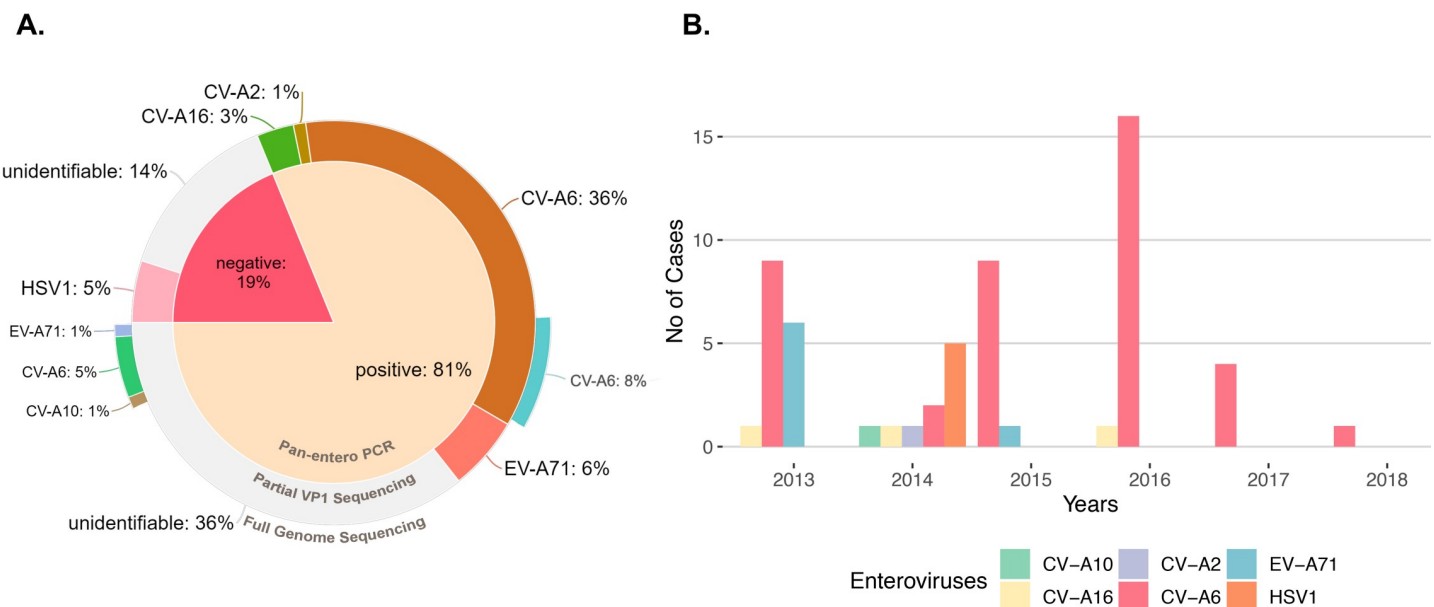

**Fig 1. CV-A6 is the major etiological agent for symptomatic HFMD in Singapore, 2013–2018. (A)** Percentage of enteroviruses detected in the study (n = 101); 81% of throat-swab samples were tested positive for presence of enteroviruses using pan-entero RT-PCR. 19% of the total cohort tested negative for presence of enterovirus. HSV1 was isolated from 5 patients (5% of total cohort) where pan-entero PCR tested negative for enteroviruses. 14% of the cohort was untypeable by partial VP1 sequencing while pan-entero PCR tested negative; 29% was unidentifiable while pan-entero PCR tested positive for enteroviruses. Virus enriched NGS was carried out on 8 known CV-A6 samples which confirmed the presence of full length CV-A6 and 7 previously un-identified samples which revealed n = 5 CV-A6, n = 1 EV-A71 and n = 1 CV-A10 infections. **(B)** Enteroviruses detected in symptomatic HFMD cohort stratified by year of collection.

40% of the clinical specimens were successfully typed to CV-A6 across all years during the period of the study, suggesting that CV-A6 is likely gaining the spotlight as the preponderant causative agent in Singapore (**Fig 1B**). Interestingly, through routine virus isolation using throat-swab samples, human herpes simplex virus type 1 was also isolated from 5 patients despite being negative for pan-entero test (**Fig 1B**) which is also confirmed by transmission electron microscopy (**S2 Fig**). The sequences were deposited into GenBank with accession numbers MW275077-MW275119.

Maximum likelihood phylogenetic trees for EV-A71, CV-A6 and CV-A16 were constructed with partial sequences from Sanger sequencing (**Fig 2**). Samples typed to EV-A71 clustered closely to the subgenogroup B4 (**Fig 2**) and samples typed to CV-A16 clustered to subgenotype B1b and B1c (**Fig 2**). Interestingly, the KK Women's & Children's Hospital samples classified into subgenogroup B4 were closely related to outbreak strains in Singapore, Malaysia and Western Australia during 2001, in which a previous study published linked epidemic activities involving EV-A71 across the Asia-Pacific region from 1997 to 2001 [20].

To compare the local CV-A6 samples collected in Singapore against those globally, we downloaded all VP1, CV-A6 sequences in GenBank and aligned against the Singapore CV-A6 sequences. Thirty out of the thirty-five CV-A6 sequences sampled from Singapore were found in a well-supported monophyletic clade (bootstrap support = 92%) of viruses that encompass strains found in Southeast Asia (Malaysia and Thailand), Asia (China, Japan and India), Europe (Denmark, U.K., Germany, Italy, Spain and Sweden) as well as the U.S (**Fig 3**). Given the disparate and highly uneven levels of global sampling (**Fig 4A and 4B**), the analyses here neither postulates that the CV-A6 viruses collected in Singapore were attributed to an overseas source from any of the aforementioned countries nor does it suggest that Singapore is the source country of these lineages of viruses found overseas. It does, however, indicate that the

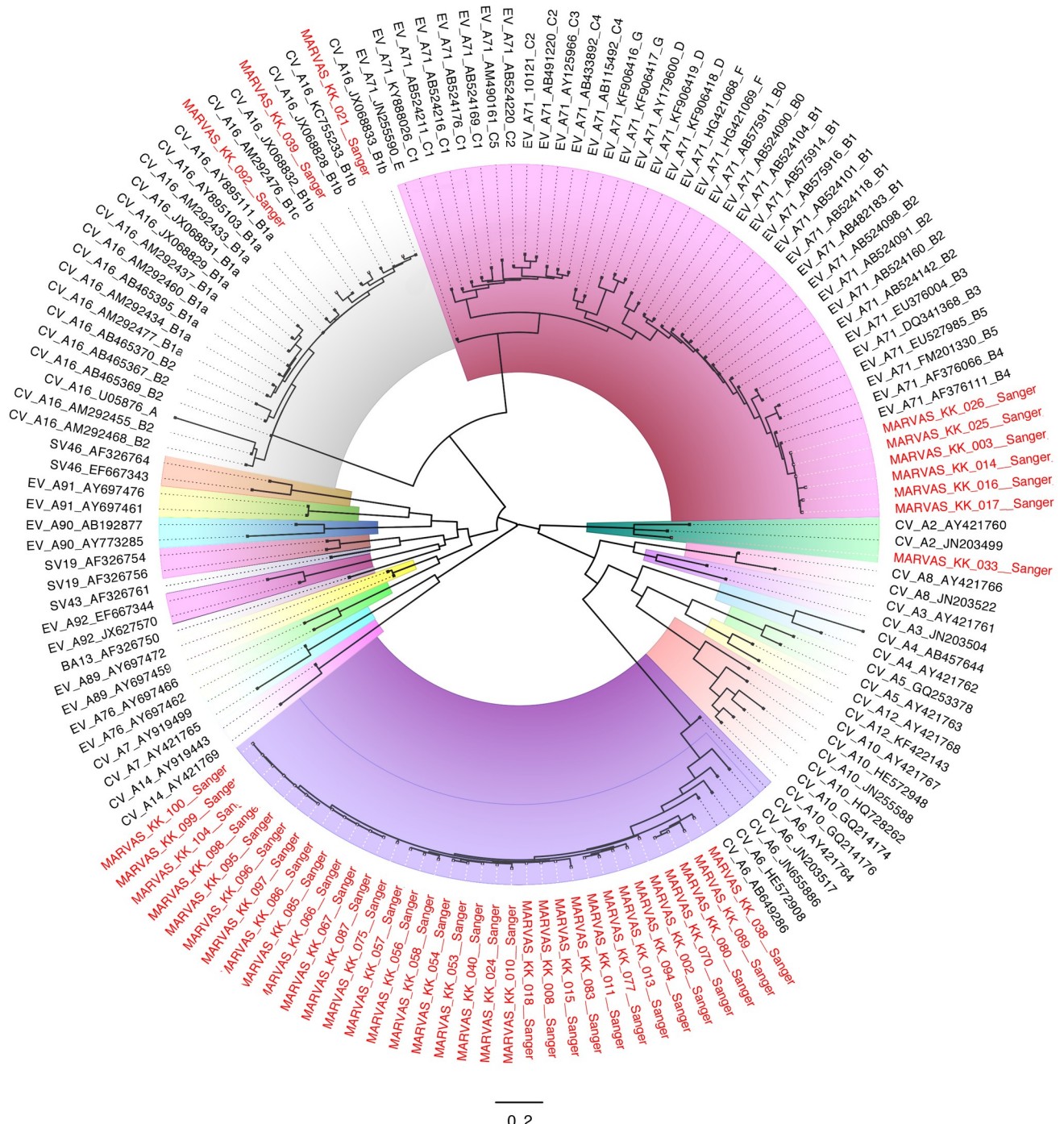

**Fig 2. Maximum likelihood phylogeny of partial VP1 sequences from HFMD symptomatic patients.** Midpoint rooted phylogenetic tree was constructed by Sanger sequencing of partial VP1 sequence using RAxML with the general reversible time substitution model and gamma-distributed rate heterogeneity. Clades were highlighted for different serotype of enteroviruses. Sequences from the symptomatic HFMD cohort were denoted with red at taxa.

dominant genetic lineage of CV-A6 viruses found in Singapore coincide with viruses that were found globally.

Furthermore, we recruited 245 healthy volunteers routinely from childcare centres across Singapore; healthy volunteers were screened for the absence of fever, rashes, blisters and oral

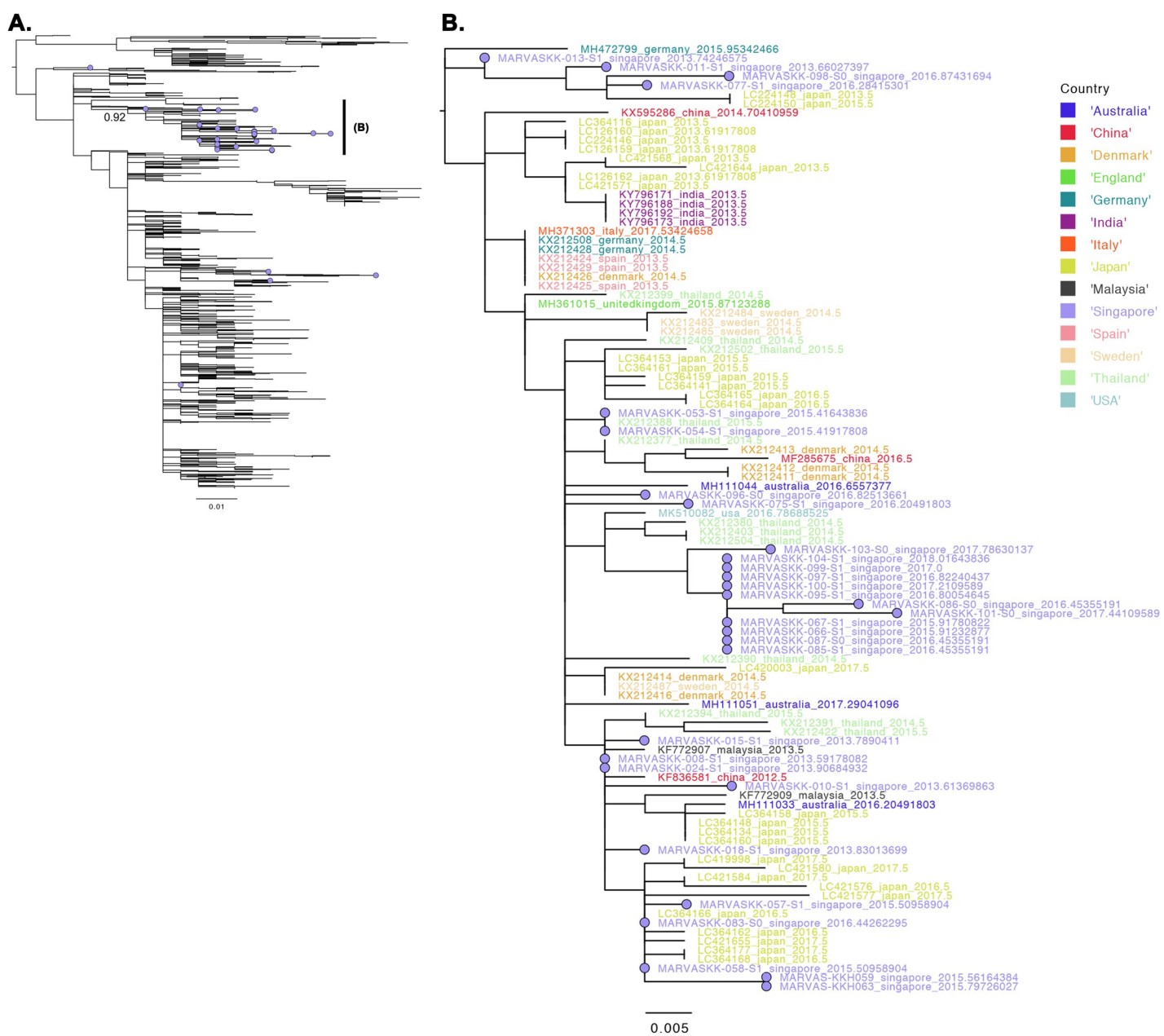

**Fig 3. Maximum likelihood tree of CV-A6 viruses.** (A) Phylogenetic tree of GenBank reference and Singaporean sequences. Singapore sequences are annotated with violet coloured tips and are largely found in the two labelled monophyletic clades of sequences with well-supported internal nodes (bootstrap support = 92%). (B) Zoom of the labelled subtrees in (A).

ulcers during sample collection. Based on recall from questionnaires issued, 43.3% (or n = 106) of children reported to be contracted with HFMD previously while 41.2% (or n = 101) of them did not suffer from HFMD prior to this study (**Fig 5A**). Therefore, the prevalence of HFMD within our study population stands at 43.3% (99% CI: 35.4%, 51.5%). Further stratification of the healthy paediatric cohort according to gender revealed that gender specific prevalence of HFMD among males and females are 40.0% (99% CI: 29.1%, 52.0%) and 46.2% (99% CI: 35.4%, 57.3%). Among those reported to have HFMD before, fever, ulcers and blisters were commonly observed during their symptomatic period, amidst other symptoms that

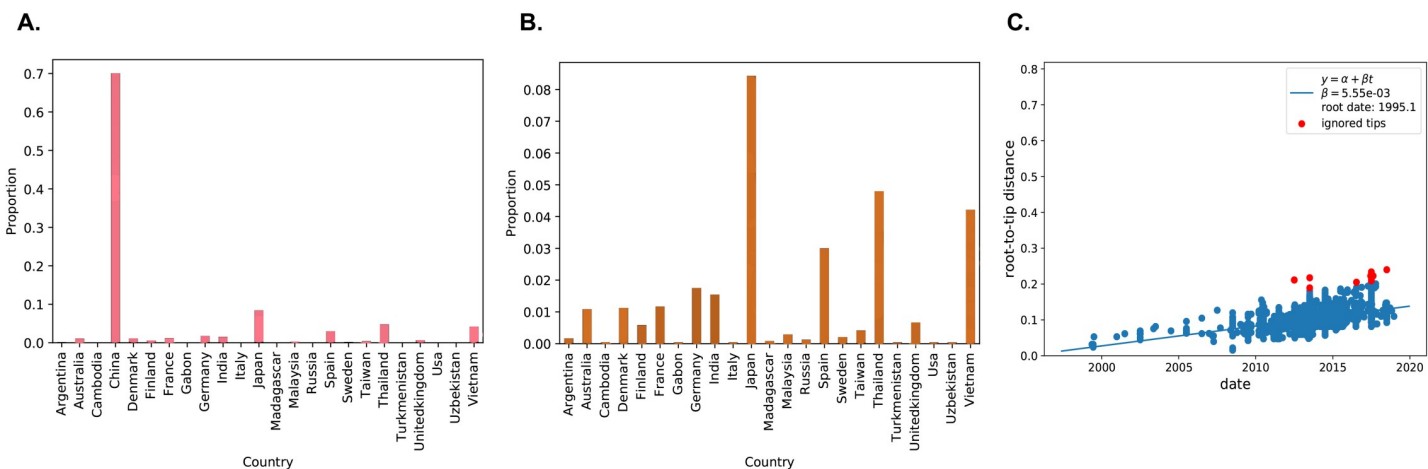

**Fig 4. Global CV-A6 phylodynamic.** Proportion of VP1 sequences of CV-A6 viruses with country and collection date information downloaded from Genbank. **(A)** including China; **(B)** excluding China. **(C)** Regression of root-to-tip genetic distances against time generated by TreeTime.

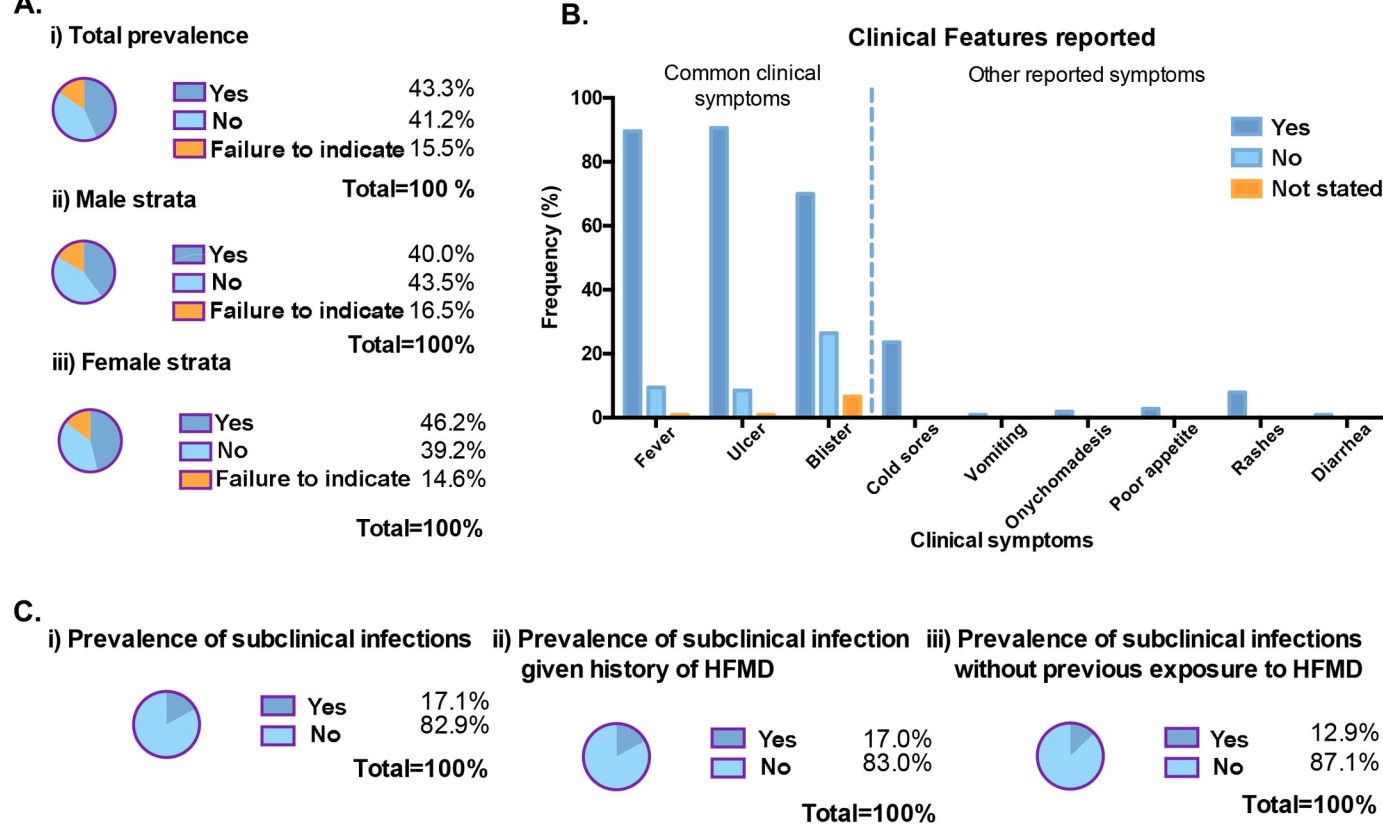

**Fig 5. Previous HFMD exposure reported from healthy pediatric volunteers, 2012–2018. (A)** The prevalence of HFMD is calculated based on recall from the questionnaires distributed to parents and/or guardians. **i)** Total prevalence of HFMD before stratification. **ii)** & **iii)** are prevalence that were stratified according to gender. **(B)** Clinical features were recorded based on recall during the period with which the child came down with HFMD. **(C)** Prevalence of subclinical infection among healthy pediatric volunteers according to the pan-entero RT-PCR assay. **i)** Overall prevalence of subclinical infections in childcare volunteers cohort while **ii)** & **iii)** are prevalence of the former stratified according to previous history of HFMD.

were also reported and recorded (**Fig 5B**). Interestingly, 2 of the children had reported having onychomadesis (shedding of nails), which is an atypical manifestation of HFMD found to be associated with CV-A6 infection[21].

Afterwards, previously mentioned 245 healthy children saliva samples were subjected to pan-entero RT-PCR assay for enterovirus detection. From our analysis, 17.1% (n = 42) were tested positive for enterovirus, placing the prevalence of subclinical infection within our study population at 17.1% (99% CI: 11.8%, 24.2%) (**Fig 5C**). Further stratification of the study population according to past exposures to HFMD revealed that prevalence of subclinical infection given a history of HFMD is 17.0% (99% CI: 9.61%, 28.2%) (**Fig 5C**). Comparatively, prevalence of subclinical infection without prior exposure to HFMD is 12.9% (99% CI: 6.54%, 23.8%); unlikely to be a risk factor associated with subclinical infection. In general, there appears to be a basal level of subclinical infection in most childcare centres (**S3 Fig**). Filtering the subclinical infections according to various age group, children ages 5 and 6 tends to have a slightly higher frequency of subclinical infection, which is possibly attributed by other parameters that were not collected in this study, such as cleaning regimen of childcare centers and households, level of physical activity and outdoor exposure, as well as personal hygienic behavioural practices (**S4 Fig**).

## Discussion

EV-A71 was one of the predominant HFMD causative agent in Asia since 1965 [22] [5]. Between 1997 and 2003, subgenogroup B4 of EV-A71 outbreaks were reported in Singapore, Malaysia, Taiwan, Australia and Japan [5]. However, subgenogroup B5 was later found to be circulating in Singapore, Malaysia, Taiwan and Japan up till late 2012, signalling a general shift in EV-A71 subgenogroup across the continent [5]. Interestingly, samples typed to EV-A71 in our symptomatic cohort were clustered under subgenogroup B4 exclusively. The presence of EV-A71 subgenogroup B4 exclusively at a lower frequency compared to other HFMD causative strains is rather alarming as there is a possibility of EV-A71 subgenogroup B4 to re-emerge after long incubation to contribute a large HFMD outbreak. As early as 1972, CV-A16 has been notoriously reported to be involved in HFMD outbreaks and epidemics in Singapore [23]. Yet again, proportionately lesser clinical samples were typed to CV-A16 in this study, which was a longstanding causative agent extensively involved in outbreaks and epidemics documented in Singapore; 3 cases of CV-A16 were detected in the symptomatic HFMD cohort and belonged to sub-genotype B1b and B1c.

CV-A6 was found to have the highest basic reproductive number when compared with other known dominant causative agents, CV-A16 and EV-A71 [24]. Since CV-A6 cases were rare before year 2000 in several HFMD endemic countries, immune naiveness of the population coupled with the high transmissibility stress that CV-A6 had the greatest epidemic potential [24]. Indeed, a previous study reported that CV-A6 became the major etiological agent of HFMD in Singapore since late 2009 overtaking EV-A71 and CV-A16 [23]. This emergence of CV-A6 led to the chain transmission of the paediatric disease and as such the number of HFMD cases snow-balled and doubled later in 2010 and further more in 2012 [23]. Increasingly, several countries in Asia have reported the rocketed rise in CV-A6 cases recently, emphasizing the need for viral surveillance particularly to guide clinical management of the disease and for public health purposes [25].

Among the children recruited into our cross sectional study, retrospective assessment of the questionnaire revealed that the prevalence of HFMD among the childcare cohort stands at 43.3% (**Table 2**). Downstream stratification did not pinpoint gender to be a risk associated with HFMD infection. Symptoms recalled during the HFMD disease period bears parallel with

**Table 2. Comparison of the socio-demographic characteristics among healthy children volunteers.** Parameters such as age, race, gender and previous HFMD history were collected based on recall from our questionnaire. A total of 245 children without signs and symptoms indicative of HFMD at the point of sample collected was recruited. Based on the Pan-Entero RT-PCR assay established previously using clinical samples, we stratified the children according to presence or absence of subclinical infection to identify possible risk factors associated with HFMD.

| | Subclinical (%) | No infection (%) |
|---|---|---|
| **Characteristics** | **(n = 42)** | **(n = 203)** |
| **Age (years)** | | |
| **3** | 8 (19.0) | 35 (17.2) |
| **4** | 6 (14.3) | 51 (25.1) |
| **5** | 12 (28.6) | 54 (26.6) |
| **6** | 13 (31.0) | 54 (26.6) |
| **Not stated** | 3 (7.1) | 9 (4.4) |
| **Race** | | |
| **Chinese** | 35 (83.3) | 162 (79.8) |
| **Malay** | 2 (4.8) | 19 (9.4) |
| **Indian** | 1 (2.4) | 6 (2.9) |
| **Eurasian** | 1 (2.4) | 1 (0.5) |
| **Others** | 3 (7.1) | 15 (7.4) |
| **Gender** | | |
| **Male** | 22 (52.4) | 93 (45.8) |
| **Female** | 20 (47.6) | 110 (54.2) |
| **Previous HFMD history** | | |
| **Yes** | 18 (42.8) | 88 (43.3) |
| **No** | 13 (31.0) | 88 (43.3) |
| **Failure to indicate** | 11 (26.2) | 27 (13.4) |

that observed by pediatric patients in the earlier portion of this study. Further stratification based on demographics did not reveal potential risk factors associated with HFMD as well. Particularly, prior exposure to HFMD will not significantly increase or decrease the risk of a subsequent case of HFMD. HFMD is caused by a plethora of viruses from the genus *Enterovirus* of the family *Picornaviridae* and prior infection with one enterovirus may not necessarily confer protective immunity against all other serotypes known to cause HFMD [4]. There are only few studies that attempts to identify risk factors associated with HFMD.

Additionally, we detected enteroviruses in a large amount of saliva samples collected routinely from childcare centres across Singapore, allowing us to identify subclinical infections. This amount of subclinical infections associated with symptomatic HFMD in childcare centres is rather alarming. It will be of interest to study the host and viral factors contributing to asymptomatic status to these individuals as well as, the long term effect of these subclinical enterovirus infection on children health, immunity and development. All in all, routine systematic surveillance programs will be essential in future to monitor and study these subclinical infections lurking in the background.

In retrospect, our findings support that CV-A6 is increasingly gaining the spotlight as the most prevalent serotype responsible for causing HFMD in Singapore from the year 2013 to 2018. Such shift in seroprevalence from 2,000s could be due to the lack of herd immunity against CV-A6, making the infection highly susceptible. This is rather alarming as in some occasions CV-A6 is associated with post-symptomatic disease complications such as desquamation and onychomadesis [26]. Geographically, regions such as China [27], Japan [25], Malaysia [28], Spain [29] and Finland [30] also reported similar shifts in the dynamics of

HFMD infection, mostly citing CV-A6 as an emerging dominant circulating agent. Perhaps a greater understanding of the disease susceptibility can better guide disease prevention and control measures, as well as appropriate clinical management against future outbreaks and HFMD epidemics.

VP1 protein of enteroviruses contains a number of neutralization domains which correspond with enterovirus serotypes and consequently the phylogenetic lineage. VP1 sequences are therefore widely used as molecular determinant of serotypes and genotypes in the field. It has been well established that VP1 and VP4 sequences of enteroviruses are well suited for serotyping and genotyping of enteroviruses [31]. Traditionally, VP1 sequencing was favoured for genotyping of enteroviruses since the development of partial VP1 sequencing and as such 333bp sequences dominated the GenBank. However, the short genomic region sequenced severely hampered our ability to make a more detailed phylodynamic and epidemiological inferences of CV-A6 viruses. There are temporal resolution limits inherent within viral genomes as the loss of information associated with such short sequences leads to decreased phylogenetic and molecular clock signals (S1 Text). Coupled with the highly uneven and inadequate sampling of CV-A6 viruses worldwide, it is currently not possible to infer transmission, migration and origins of these viruses [32]. Our effort to determine the temporal event of CV-A6 genetic shifts in the past with global CV-A6 strains were rather futile as most of the CV-A6 sequences in GeneBank were just partial VP1 sequences yielding a highly diffuse temporal signal (Fig 4C). This strongly highlight a need for full genome sequences of enteroviruses going forward as much information is lost by partial gene sequencing, rendering the phylodynamic studies rather impractical. On the other hand, silent enterovirus infections are prevailing among healthy children volunteers recruited into this study; we found that children can be subclinically infected without symptoms indicative of HFMD; this clearly emphasizes the need for public health intervention for the control and spread of symptomatic and subclinical HFMD, especially in Singapore where HFMD is endemic and the increasing adoption of dual-income family structures that requires children to be put under the care and attention of childcare centres.

## Supporting information

**S1 Text. Method for temporal analysis of CV-A6 strains**
(DOCX)

**S1 Table. Distribution of viruses typed from Partial VP1 detection assay.** Clinical samples were typed using primers that partially amplified the VP1 region of the viral genome and sequencing results produced the following distribution, with most of the samples being typed of CV-A6, followed by EV-A71 and lastly, CV-A16. diagnostic assay to detect for enterovirus. Enterovirus is detected in 71% and 87% of the saliva and throat swab samples respectively. As such, throat swab is comparatively more sensitive for detection and diagnoses.
(XLSX)

**S1 Fig. Clinical symptoms presented by pediatric patients admitted due to HFMD.** These clinical symptoms were recorded from the pediatric patients during admission and several symptoms were presented by the majority of the patients, including fever, mouth ulcer, skin lesion, rashes and papules.
(TIF)

**S2 Fig. Transmission Electron Micrograph of Herpes Simplex Virus 1 isolated from MAR-VAS_KK_029.** Isolated HSV1 viruses were used to infect RD cell at MOI of 1. Infected cells

were fixed at 16 hours post infection and processed for TEM analysis.
(TIF)

**S3 Fig. Distribution of subclinical infection across all childcare centers sampled in this study.** Based on the Pan-Entero RT-PCR assay, the subclinical infections were dispersed across all the childcare center sampled, revealing a basal level of subclinical infection across most childcare centers.
(TIF)

**S4 Fig. Distribution of subclinical infection across various age groups.** Subclinical infections were stratified according to the different ages of the children.
(TIF)

## Acknowledgments

We would like to acknowledge individuals not mentioned as co-authors from the clinical team of KK Women's and Children's Hospital for collecting samples and suggestions. We would also like to thank Dr Patchara Phuektes for all of the help while the project was starting out. We will also like to thank Regina Lee Ching Hua, Dr Fiona Mei Shan Teo, June Guat Nee Koh, Mariya D/O Parimelalagan, Dr Parveen Kaur, Shirley Lam, Kai Zhi Wong and Florentino Gemuel Paul Villar for their support and help along the project.

## Author Contributions

**Conceptualization:** Chia Yin Chong, Justin Jang Hann Chu.

**Funding acquisition:** Justin Jang Hann Chu.

**Investigation:** Nyo Min, Yasmin Hui Binn Ong, Alvin X. Han, Si Xian Ho.

**Resources:** Emmerie Wong Phaik Yen, Kenneth Hon Kim Ban, Sebastian Maurer-Stroh.

**Supervision:** Chia Yin Chong, Justin Jang Hann Chu.

**Writing – original draft:** Nyo Min, Yasmin Hui Binn Ong, Alvin X. Han.

**Writing – review & editing:** Nyo Min, Justin Jang Hann Chu.

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
