## [Decision Letter · Decision Letter 0]

1 Sep 2020

Dear Dr. Chu,

Thank you very much for submitting your manuscript "Epidemiology of Hand Foot and Mouth Disease in Paediatric Patients and in Community: A Retrospective Cohort Study, 2013-2018" for consideration at PLOS Neglected Tropical Diseases. As with all papers reviewed by the journal, your manuscript was reviewed by members of the editorial board and by several independent reviewers. The reviewers appreciated the attention to an important topic. Based on the reviews, we are likely to accept this manuscript for publication, providing that you modify the manuscript according to the review recommendations. 

Please address all reviewers’ and the following comments:

1. Please improve the clarity of the figures and increase the font size of the labels/text.

2. Methods, please clarify how the patients were recruited. Were there only 104 HFMD patients over 5 years in the KK Women’s and Children’s Hospital?

3. Tables 1 & 2 and Figure 5, please change “TRUE”/”FALSE” to “Yes”, “No”

4. Discuss the use of VP1 sequence on the interpretation of results

Sincerely,

Eric HY Lau, Ph.D.

Associate Editor

Pedro Vasconcelos

Deputy Editor

Please address all reviewers’ and the following comments:

1. Please improve the clarity of the figures and increase the font size of the labels/text.

2. Methods, please clarify how the patients were recruited. Were there only 104 HFMD patients over 5 years in the KK Women’s and Children’s Hospital?

3. Tables 1 & 2 and Figure 5, please change “TRUE”/”FALSE” to “Yes”, “No”

4. Discuss the use of VP1 sequence on the interpretation of results

Reviewer's Responses to Questions

**Key Review Criteria Required for Acceptance?**

**Methods**

-Are the objectives of the study clearly articulated with a clear testable hypothesis stated?

-Is the study design appropriate to address the stated objectives?

-Is the population clearly described and appropriate for the hypothesis being tested?

-Is the sample size sufficient to ensure adequate power to address the hypothesis being tested?

-Were correct statistical analysis used to support conclusions?

-Are there concerns about ethical or regulatory requirements being met?

Reviewer #1: This is a retrospective studies for the surveilance study of HFMD in Singapore. 

The enrolled patient was appropriate for this objective.

Reviewer #2: Since authors have just describe the genotype of Enterovirus for HFMD. The term of "cohort" in Methods is not right. Actually, authors have not described the design following the guideline of cohort study. The design of the present study is open to question. We could take it as a surveillance for HFMD epidemic in Singapore. Considered that, it is better to rewrite the Methods. I am confused the description of the present study. More work is need to make it clear.

Reviewer #3: (No Response)

**Results**

-Does the analysis presented match the analysis plan?

-Are the results clearly and completely presented?

-Are the figures (Tables, Images) of sufficient quality for clarity?

Reviewer #1: They genotyped 101 HFMD infected samples and demonstrated that CV-A6 is the major etiological virus for the outbreaks during 2013 to 2018 in Singapore. 

The short sequence of VP1 was used to compare the local CV-A6 samples in Singapore against those globally, however, the dominant genetic lineage of CV-A6 viruses found in Singapore coincide with viruses that were found globally. The conclusion is not clear.

All of the figures are complex and not easy to understand

Reviewer #2: 1.Only over 100 samples collected from 2013 to 2018, why did authors mentioned "recent outbreaks" in Abstract?

2.The characteristics of patients and health controls should be shown in one table. Are there any change of them from 2013 to 2018?

3.Figure 5 is questionable. Why are the data from 2012 included in the figure? What is the meaning of Figure 5B? Is it the data from 2010 to 2018? Are there any time trend for it? Figure 5A and 5C could be shown in a table.

Reviewer #3: In page 8 line 155, understandably a full recovery means no symptoms and fever present. However, does the partial recovery status mean that some of the symptoms such as mouth ulcers or rashes are still present or just a slight fever? 

Page 22. Please clarify the grading for characteristics in Table 2. 

Page 27 Figure 3. The text in this figure is difficult to distinguish, even in the PDF file. Is there a clearer figure to replace this one?

**Conclusions**

-Are the conclusions supported by the data presented?

-Are the limitations of analysis clearly described?

-Do the authors discuss how these data can be helpful to advance our understanding of the topic under study?

-Is public health relevance addressed?

Reviewer #1: The conclusion is clear and well addressed.

The authors should try to determine the temporal event of CV-A6 genetic shifts in the past while comparing the local sequence with global strains, however, as most of the CV-A6 sequences in GeneBank were only partial VP1 sequences leading to a highly diffuse temporal signal.

The author raise the silent Enterovirus infections in this study, indicating that spread of subclinical HFMD is happened.

Reviewer #2: The conclusion seems to be supported by the data presented. But we need more details from the study.

Reviewer #3: (No Response)

**Editorial and Data Presentation Modifications?**

Reviewer #1: For Abstract: “We studied symptomatic paediatric HFMD cases from 2013 to 2018 in Singapore. Surveillance for subclinical enterovirus infections was also performed in childcares at the same time period.” At least, please describe how to detect the different types of Enteroviruses and sequence and Phylogenetic analysis.

Reviewer #2: (No Response)

Reviewer #3: Page 9 line 188. A typo was written as ‘throat-swan’.

Page 10 line 194. The abbreviation for CV-A16 was used inconsistently and was written as CA16.

Page 11 Line 225. Reference style inconsistency.

**Summary and General Comments**

Reviewer #1: In this study, Min et al., reported that CV-A6 is the major etiological 38 agents for the recent HFMD outbreaks during 2013 to 2018 after genotyping 101 symptomatic HFMD samples in Singapore. They also detected sub-clinical enterovirus infections in childcare centres and revealed that 17.1% (n=245) tested positive for enterovirus in saliva, without HFMD indicative symptoms at the same period. This is an interesting and important information for the Enteroviruses surveillance in Singapore. Also, the results indicate the subclinical cases in Singapore and HFMD outbreaks often occurs sporadically and frequently in the childcare and other children’s learning institutions. 

There is only minor revision should be addressed:

Keywords: “Singapore” should not be as a keyword

As for Figure 1A, I notice that there is CA-6 (36%, brown color) shown in the positive region and addition CA-6 (8% in blue color) shown in the overlap region, can the authors explain this result?

Line 307-308: “VP1 sequencing was favoured for genotyping of enteroviruses since the development for partial VP1 sequencing primers and as such 333bp sequences dominated the GenBank”, please revised the sentence as the following:

Due to the development of some VP1 sequencing primers and the 333bp sequence dominating GenBank, VP1 sequencing is favored in enterovirus genotyping..

Reviewer #2: Since authors want to describe the change by years, more work is need to illustrate that. I do not think the present study has been well reported. The desing and data presented are also questionable. More work are need to make the outline clear.

Reviewer #3: The authors of this manuscript studied HFMD and subclinical cases from the years 2013 to 2018 in Singapore. Their findings showed that majority of the samples collected were cased by coxsackievirus A6, while a smaller percentage being caused by EV-A71, CV-A16, CV-A2, and CV-A10. They also compared the positive testing rate between samples collected via throat swab and from saliva. The findings indicated that samples from throat swabs had a higher positive rate in virus detection compared to saliva samples. The authors also reported that 17.1% tested positive for enterovirus in saliva among healthy children in daycare centers.

PLOS authors have the option to publish the peer review history of their article (what does this mean?). If published, this will include your full peer review and any attached files.

Reviewer #1: No

Reviewer #2: Yes: Sheng Wei

Reviewer #3: Yes: Jen-Ren Wang
---

## [Decision Letter · Decision Letter 1]

13 Oct 2020

Dear Dr. Chu,

We are pleased to inform you that your manuscript 'An Epidemiological Surveillance of Hand Foot and Mouth Disease in Paediatric Patients and in Community: A Singapore Retrospective Cohort Study, 2013-2018' has been provisionally accepted for publication in PLOS Neglected Tropical Diseases.

Best regards,

Eric HY Lau, Ph.D.

Associate Editor

Pedro Vasconcelos

Deputy Editor

Reviewer's Responses to Questions

**Key Review Criteria Required for Acceptance?**

**Methods**

-Are the objectives of the study clearly articulated with a clear testable hypothesis stated?

-Is the study design appropriate to address the stated objectives?

-Is the population clearly described and appropriate for the hypothesis being tested?

-Is the sample size sufficient to ensure adequate power to address the hypothesis being tested?

-Were correct statistical analysis used to support conclusions?

-Are there concerns about ethical or regulatory requirements being met?

Reviewer #2: Yes, the revised method is clear and reasonable.

Reviewer #3: (No Response)

**Results**

-Does the analysis presented match the analysis plan?

-Are the results clearly and completely presented?

-Are the figures (Tables, Images) of sufficient quality for clarity?

Reviewer #2: Yee, the results are clearly and completely presented.

Reviewer #3: (No Response)

**Conclusions**

-Are the conclusions supported by the data presented?

-Are the limitations of analysis clearly described?

-Do the authors discuss how these data can be helpful to advance our understanding of the topic under study?

-Is public health relevance addressed?

Reviewer #2: THe conclusions is clear and appropriate.

Reviewer #3: (No Response)

**Editorial and Data Presentation Modifications?**

Reviewer #2: No more modificaition is needed.

Reviewer #3: (No Response)

**Summary and General Comments**

Reviewer #2: The revised manuscript is well written. I suggest it could be accepted to publication.

Reviewer #3: All comments were clarified and fixed. I have no further comments.

PLOS authors have the option to publish the peer review history of their article (what does this mean?). If published, this will include your full peer review and any attached files.

Reviewer #2: No

Reviewer #3: **Yes: **Jen-Ren Wang

---

## [Editor Report · Acceptance letter]

5 Feb 2021

Dear Dr. Chu,

We are delighted to inform you that your manuscript, "An Epidemiological Surveillance of Hand Foot and Mouth Disease in Paediatric Patients and in Community: A Singapore Retrospective Cohort Study, 2013-2018," has been formally accepted for publication in PLOS Neglected Tropical Diseases.

Best regards,

Shaden Kamhawi

co-Editor-in-Chief

Paul Brindley

co-Editor-in-Chief
